# Increasing countries' financial resilience through global catastrophe risk pooling

Alessio Ciullo [1,2] ✉, Eric Strobl[3], Simona Meiler [1,2], Olivia Martius[4] & David N. Bresch [1,2]

Extreme weather events can severely impact national economies, leading the recovery of low- to middle-income countries to become reliant on foreign financial aid. Foreign aid is, however, slow and uncertain. Therefore, the Sendai Framework and the Paris Agreement advocate for more resilient financial instruments like sovereign catastrophe risk pools. Existing pools, however, might not fully exploit their financial resilience potential because they were not designed to maximize risk diversification and because they pool risk only regionally. Here we introduce a method that forms pools by maximizing risk diversification and apply it to assess the benefits of global pooling compared to regional pooling. We find that global pooling always provides a higher risk diversification, it better distributes countries' risk shares in the pool's risk and it increases the number of countries profiting from risk pooling. Optimal global pooling could provide a diversification increase to existing pools of up to 65 %.

Extreme weather events like tropical cyclones, floods, and heavy precipitation can have severe impacts on economies, leading to a short-term deterioration of several macro-economic variables. In the Caribbean region, for example, an average hurricane strike was found to cause an annual growth loss of about 0.84%[1], a local income growth loss of 1.5%[2], a total tax revenue loss of 5.3%[3], a multifold increase in monthly average inflation[4], and an appreciation of real exchange[5].

These deteriorated macro-economic scenarios are likely to require increases in government spendings[6] via short-term deficit financing, which in turn leads to debt increase[3]. For countries facing pre-existing debt sustainability issues this may be very costly[7] and, therefore, their recovery often relies on financial aid from international donors acting as insurers of last resort. Although foreign financial aid can help mitigate the effect of natural disasters on economic growth[8], it is also generally considered to be a slow and uncertain *ex-post* financial instrument[9]. Foreign financial aid may take months to materialize and it is impossible to assess a priori what amount, if any, will be provided and under what conditions. Historically, only about 60% of the humanitarian requests are covered and funds have not been equally allocated between emergencies[10,11]. In contrast, *ex-ante*

financial instruments, e.g., insurance, provide faster and more predictable funding flows in the aftermath of disasters and allow governments to spread costs over time at a predictable rate[10]. Furthermore, *ex-ante* financial instruments complement non-financial disaster risk management strategies as they may foster investments in risk reduction and increase preparedness and adaptation[11].

Several international high-level policy agendas advocate for strengthening financial resilience towards the impact of extreme natural hazards via *ex-ante* financial instruments[12]. For instance, the 2015 Sendai Framework for Disaster Risk Reduction promoted by the United Nations outlines four actions to prevent and reduce disaster risk. In this regard, the framework's third priority stress the importance of *ex-ante* mechanisms such as insurance to reduce financial impacts of disasters on governments[13]. Also, Article 8 of the Paris Agreement reaffirmed the Warsaw International Mechanism for Loss and Damage and recognizes risk insurance facilities as effective instruments to limit the impacts of extreme weather events[14]. Following these calls, the *InsuResilience* Global Partnership[15] was launched by the G20 and V20 Groups at COP23 in November 2017. *InsuResilience* identifies sovereign catastrophe risk pools, a financial mechanism where different

[1]Institute for Environmental Decisions, ETH Zurich, Zurich, Switzerland. [2]Swiss Federal Office of Meteorology and Climatology MeteoSwiss, Zurich, Switzerland. [3]Department of Economics and Oeschger Centre for Climate Change Research, University of Bern, Bern, Switzerland. [4]Institute of Geography and Oeschger Centre for Climate Change Research, University of Bern, Bern, Switzerland. ✉e-mail: alessio.ciullo@usys.ethz.ch

countries pool their risk into a single portfolio, as being a promising *ex-ante* instrument, especially for countries with low geographical (e.g., due to a limited size) or temporal (e.g., due to a limited borrowing capacity) risk spreading potential[9].

An effective risk pooling makes countries' shares of the pool's risk lower than their individual risks[16] and, therefore, it lowers countries' technical premiums compared to when they buy insurance separately. In particular, the technical premium is mainly determined by three factors: operational costs, cost of capital and annual expected losses[17]. Risk pooling reduces operational costs and the cost of capital. Operational costs are reduced because they are shared among all countries in the pool thus enabling economies of scale. Capital costs reduction provides the largest premium reduction and it is achieved via increased financial efficiency[11,17], which is in turn reached primarily via increased risk diversification. Risk diversification relies on the idea that large losses will not be experienced by all countries simultaneously. Therefore, insuring the pooled risk requires much less capital than insuring all individual risks separately[10,18]. Financial efficiency is also increased via the establishment of joint reserves. These allow retaining a larger risk share than what countries could individually retain, thus reducing the fraction of risk transferred to the reinsurance market and the associated costs. Furthermore, a reduction in the costs of reinsurance is achieved through larger excess risk transactions to the reinsurance market.

Currently, three sovereign catastrophe risk pools exist: the Caribbean Catastrophe Risk Insurance Facility (CCRIF), the African Risk Capacity (ARC), and the Pacific Catastrophe Risk Assessment and Financing Initiative (PCRAFI). CCRIF and PCRAFI cover tropical cyclones, excess rainfall and seismic risks; ARC covers mainly drought risk and, for few countries, also tropical cyclone and flood risk. While these pools provide significant benefits to their members, they also suffer from various weaknesses. First, foreign financial aid may be still required since the three pools provide coverage that is sufficient only for a first response and not a full recovery. Additionally, members may choose not to purchase sufficient coverage in order to lower premium costs. Moreover, some members in PCRAFI and ARC still rely on foreign donors to pay their premium. Finally, pools' risk diversification might be limited since pools were designed to serve the interest of individual members without focusing on diversification aspects and they pool risk only regionally, thus missing the potential benefits of including countries located elsewhere (World Bank[11]). The present paper focuses on this last issue.

In the paper, we introduce a method to find *optimal* risk pools, i.e., those with the highest risk diversification achieved with the least number of countries, and appyl it to assess and compare risk diversification benefits stemming from regional and global optimal pooling of tropical cyclone risk. We first identify the hypothetical optimal regional pools for four regions prone to tropical cyclones and assess to what extent global pooling might improve their risk diversification. We then focus on the two existing regional pools covering tropical cyclone risk, i.e., CCRIF and PCRAFI, to assess their current risk diversification and the extent to which they might benefit from regional and global optimal pooling.

## Results

We identify four geographical regions prone to tropical cyclones: East Asia & Pacific (EAP), Latin America & Caribbean (LAC), South Asia (SA) and Sub-Saharan Africa (SSA) (see also Supplementary Fig. S1). The EAP region comprises 26 countries, the LAC region 38, the SSA region 16 and the SA region only 7. Regions are identified following the World Bank's official regional classification[19] while retaining only middle- to low-income countries facing tropical cyclone risk.

A 10000-year series of total annual tropical cyclone losses is reconstructed to assess risk diversification of sovereign catastrophe pools (*pools* for short hereafter) (see *Method*). The pools' risk diversification is assessed considering the 200-year event, which implies an $\alpha$ of 0.995 when calculating the Value-at-Risk, *VaR*, the Expected Shortfall, *ES*, and the Marginal Expected Shortfall, *MES* (see *Method*).

Hereafter, when reporting correlations of losses between countries, these refer to the yearly total losses higher than the 200-year loss and they are calculated using the Pearson correlation coefficient. Countries are reported via their ISO 3166-1 alpha-3 codes and the reader is referred to Supplementary Tables S1–S4 to match countries' ISO codes with their official names.

### Regional optimal pools

Finding the optimal regional pools for each of the four regions requires carrying out the first optimization step introduced in *Method* for one pool at a time, thus solving four single-objective optimization problems. The optimal pool in the LAC region has the highest diversification (0.75), followed by those in the EAP (0.66), SSA (0.5) and SA (0.33) regions (Fig. 1a). Risk diversification potentials are thus higher when more countries can join the pool.

A pool's risk diversification and composition depend on the countries' correlation structure (Fig. 1b–e). The optimal pools primarily consist of uncorrelated or poorly correlated countries within a region. This stems from obvious risk diversification considerations, as highly correlated countries are likely to experience losses simultaneously, thus decreasing the pool's risk diversification. For example, in LAC, the region which exhibits the highest intra-regional correlations, countries like Anguilla (AIA), Saint-Barthélemy (BLM), Saint Martin (MAF) and Sint Maarten (SXM) have high bilateral correlations ranging from 0.85 (AIA and BLM) to 0.95 (MAF and SXM, and MAF and BLM) and they are left out from the optimal pool. The same applies to Saint Kitts and Nevis (KNA) and Montserrat (MSR), which have a bilateral correlation of 0.75. Similar considerations can be made for the other regions, where Viet Nam (VNM) and Cambodia (KHM) in EAP, Bhutan (BTN) and Bangladesh (BGD) in SA, Zimbabwe (ZWE) and South Africa (ZAF) or Somalia (SOM) and Ethiopia (ETH) in SSA exhibit the highest bilateral correlations within their region and they are not part of the respective regional optimal pool. All these high correlations are explained by the countries' geographical proximity.

Correlations among countries alone do not fully explain the pools' composition, as this also depends on the share of countries' individual risk contributing to the pools' overall risk (see *Method*). In LAC, for example, Barbados (BRB) and Saint Lucia (LCA) have a relatively high bilateral correlation (0.54) and they are both part of the optimal regional pool. Similarly, in EAP, Samoa (WSM) and American Samoa (ASM) both belong to the optimal regional pool and they have a bilateral correlation of 0.30 (Fig. 1f–i). These countries are part of the pool because their share of individual risk contributing to the optimal pool's risk is very low (0.12 for BRB and 0.15 for LCA, 0.06 for WSM, 0.03 for ASM). In contrast, some countries, e.g., Panama (PAN), are left out from the optimal pool because they are correlated with other countries, e.g., Colombia (COL), that are part of the pool and contribute with a high individual risk share to the pool's risk (COL's risk share in LAC is 0.5, namely the highest among all countries in the pool).

### Globally diversified regional optimal pools

After finding the optimal regional pools, we explore whether—and to what extent—possible global expansions of these pools increase their risk diversification. In doing so, the search for new countries that could join an optimal regional pool is global and no longer limited to a given region. Any country not previously included in the optimal pool of its own region may join any—but only one—of the globally expanded regional optimal pools. Thus, it follows that optimal global pooling needs to be carried out simultaneously for the four regional pools solving a four-objectives optimization problem (see *Method*).

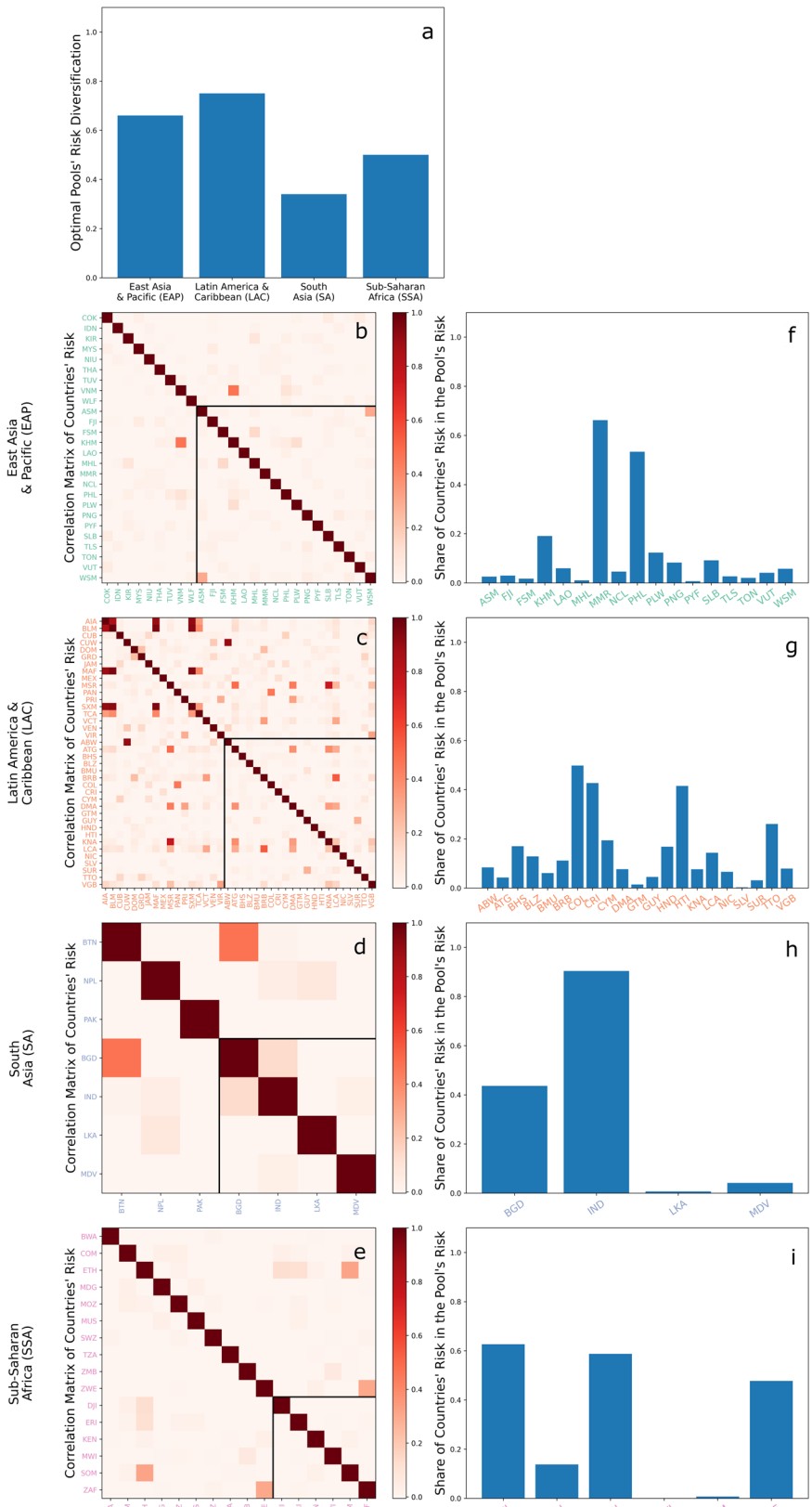

**Fig. 1 | Results for the optimal regional pools in the East Asia & Pacific (EAP), Latin America & Caribbean (LAC), South Asia (SA) or Sub-Saharan Africa (SSA) regions.** **a** shows risk diversification of the four regional optimal pools. **b**–**e** show correlation matrices and the share of countries' risk contributing to the pool's risk within each region. The correlation matrixes show the Pearson correlation coefficient for impacts with a return time of 200-y or higher for all countries in the region (full matrix) and for countries that are part of the optimal pool (sub-matrix delimited by the black line). Bar plots in **f**–**i** show shares of countries' risks contributing to optimal regional pools' risks. Countries are reported with their ISO 3166-1 alpha-3 codes, and they are colored light green, orange, light blue or pink if they respectively belong to the EAP, LAC, SA or SSA region.

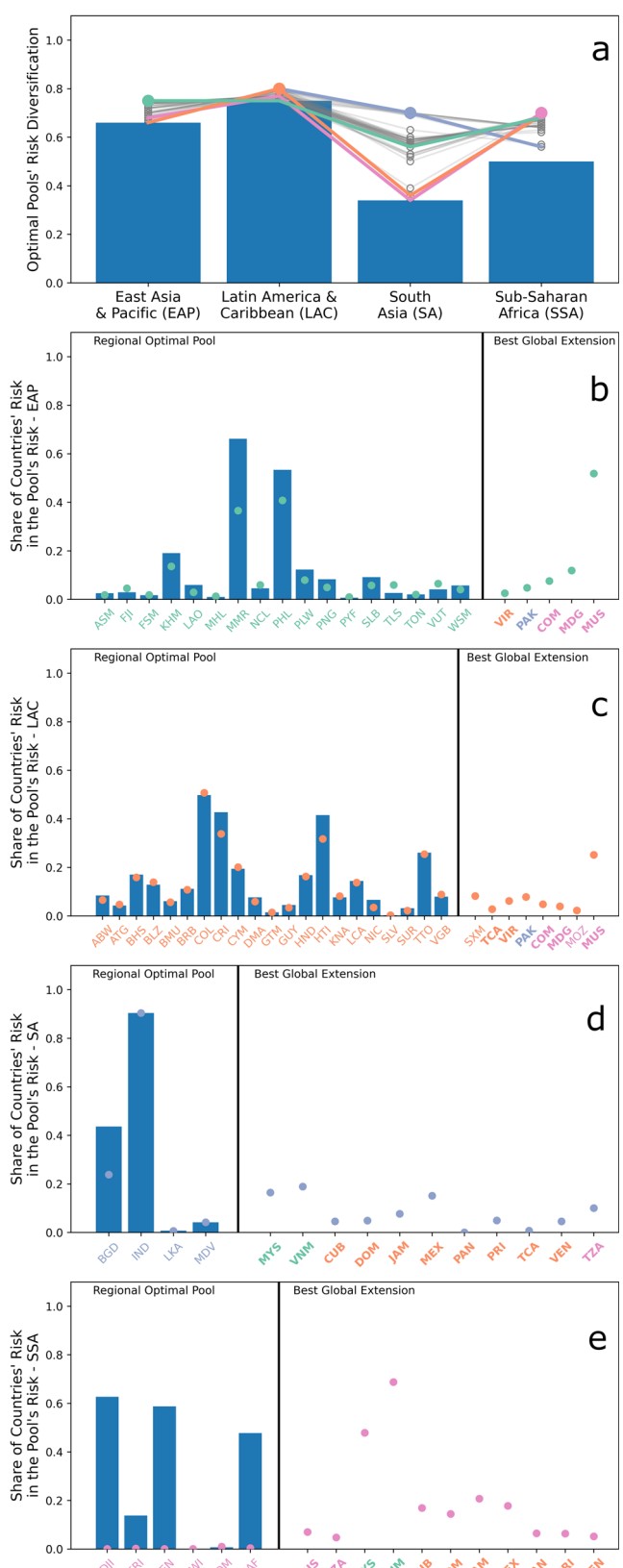

**Fig. 2 | Results for the globally diversified optimal regional pools for the East Asia & Pacific (EAP), Latin America & Caribbean (LAC), South Asia (SA) and Sub-Saharan Africa (SSA) regions. a** shows risk diversifications of the four regional optimal pools (bars) and the Pareto optimal configurations of the globally diversified regional optimal pools (continuous lines). For the latter, all configurations are reported in gray and the best configuration for each region is highlighted in light green, orange, light blue or pink if it refers to the EAP, LAC, SA, or SSA region, respectively. The highest diversification for each region is indicated with a dot following the same coloring scheme. **b–e** show, for each region, the share of countries' risk contributing to the regional optimal pool's risk (bars) and the best globally diversified optimal regional pool's risk (dots). Countries are reported with their ISO 3166-1 alpha-3 codes following the aforementioned coloring scheme. ISO codes reported in bold indicate countries that are present in more than one of the globally diversified optimal regional pool.

and SSA, benefit the most from global pooling. More precisely, the highest achievable diversification via global pooling doubles for SA (from 0.34 to 0.7) and reaches a 40% increase for SSA (from 0.5 to 0.7). In EAP and LAC, where optimal regional diversification was already high, the diversification increase is less prominent, and it amounts to a maximum of about 15% for EAP (from 0.66 to 0.75) and about 6.5% for LAC (from 0.75 to 0.8). Therefore, the four optimal regional pools reach comparable maximum risk diversifications after global pooling.

However, the maximum risk diversification is not achievable for all four pools together as trade-offs exist among the various Pareto optimal configurations of the four globally extended regional pools. The trade-off is particularly relevant for SA and SSA, as SA reaches the highest diversification when the one of SSA is lowest. Such a trade-off exists becasue some countries, i.e., Malaysia (MYS), Viet Nam (VNM), Cuba (CUB), Dominican Republic (DOM), Jamaica (JAM), Mexico (MEX), Panama (PAN), and Tanzania (TZA), are part of the best globally extended regional pool of both regions.

Overall, global pooling tends to decrease all countries' risk shares contributing to the pool's risk, and this happens because the pool's risk is redistributed elsewhere across the globe (Fig. 2b–e). Interestingly, global pooling also allows some regions, e.g., SSA and LAC, to pool countries within their own region that were not previously selected in the optimal regional pooling. This occurs because global pooling decreases the risk share of these countries in the pool's risk and thus allows them to join their own regional pool effectively. This happens even with correlated countries like Sint Maarten (SXM) and Turks and Caicos Islands (TCA), which are both part of the globally diversified LAC pool with a very low risk share (0.09 for SXM and 0.03 for TCA) despite a moderate bilateral correlation (0.35).

## Regional and global optimal diversification of PCRAFI and CCRIF

After applying the method to find hypothetical optimal regional pools and assess the effect of optimal global pooling on their risk diversification, we now focus on the two existing pools that provide coverage for tropical cyclone risk: PCRAFI and CCRIF. We assess their current risk diversification and explore to what extent regional and global optimal expansions of these pools increase their risk diversification.

Optimal regional pooling leads to a diversification increase of 35% for PCRAFI (from 0.49 to 0.66) and of about 40% for CCRIF (from 0.48 to 0.67) (Fig. 3a). In the case of PCRAFI, a diversification of 0.66 is the maximum that can be achieved since it equals the one of the optimal regional pool in the EAP region (see previous sections). For CCRIF, on the contrary, the achieved risk diversification via optimal regional pooling is about 89% of the maximum possible diversification in the LAC region. This implies that the initial design of CCRIF prevents the exploitation of the full regional diversification potential.

In terms of individual countries' share of risk contributing to the pool's risk (Fig. 3b–e), most countries in both PCRAFI and CCRIF have

Various Pareto optimal configurations of the four globally extended regional optimal pools exist (Fig. 2a). All these configurations increase risk diversification for all four pools, implying that global pooling leads to a strong Pareto improvement of the regional optimal pools. Yet, the magnitude of such an increase differs across regions. Regions where optimal regional diversification was the lowest, i.e., SA

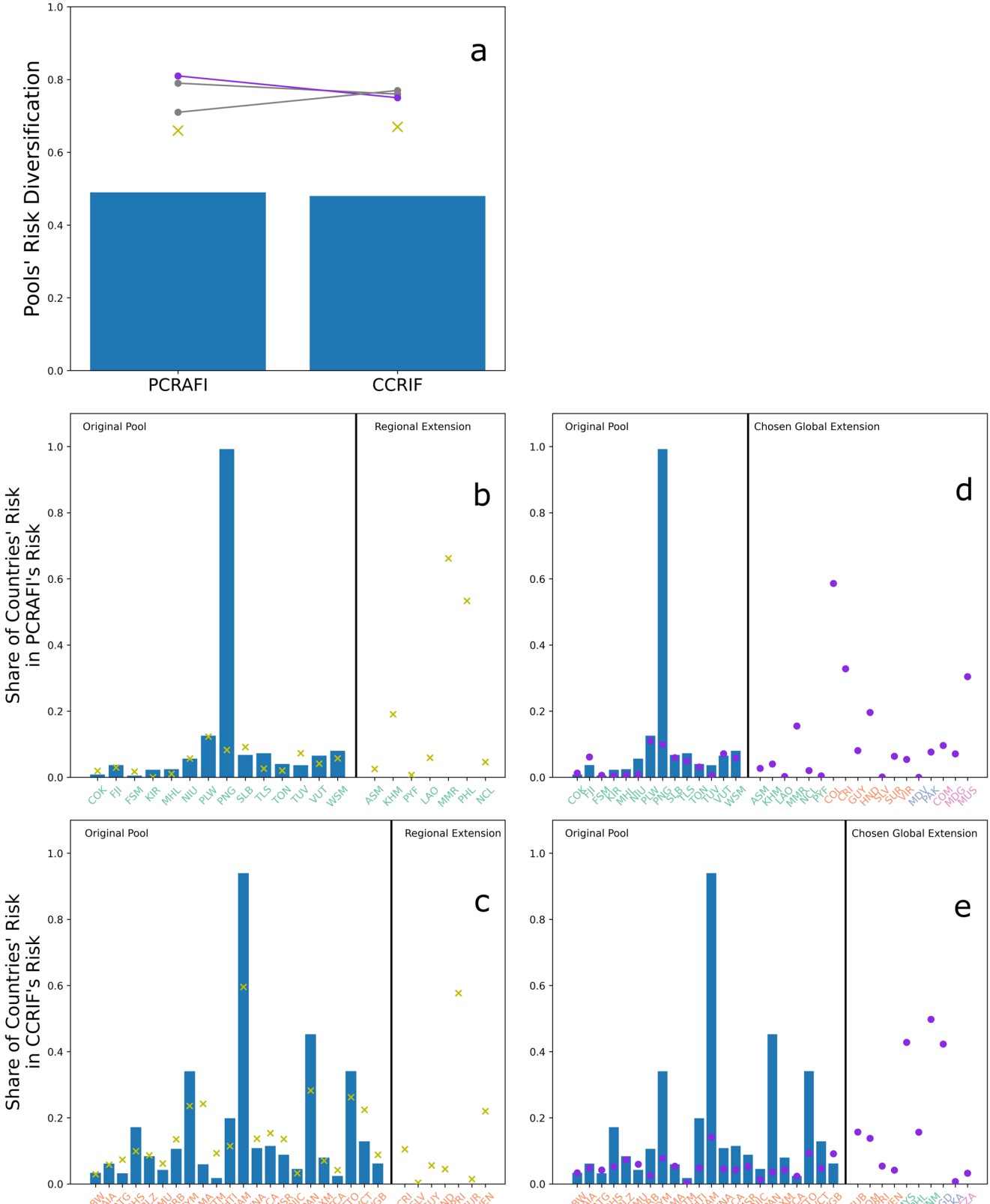

**Fig. 3 | Results of the regional and global optimal extensions of the Pacific Catastrophe Risk Assessment and Financing Initiative (PCRAFI) and the Caribbean Catastrophe Risk Insurance Facility (CCRIF). a** shows risk diversifications of the original pools (bars), the regionally (yellow cross) and globally (solid lines) diversified pools. Regarding the latter, all Pareto optimal configurations are reported in gray and the selected one (namely the one leading to the highest diversification for PCRAFI) is highlighted in purple. **b–e** show the shares of countries' risk contributing to the original PCRAFI's (**b–d**) and CCRIF's (**c–e**) risks and to their regionally (**b–c**) and globally (**d–e**) diversified pool's risks. Countries are reported with their ISO 3166-1 alpha-3 codes, and they are colored light green, orange, light blue or pink if they respectively belong to the East Asia & Pacific (EAP), Latin America & Caribbean (LAC), South Asia (SA) or Sub-Saharan Africa (SSA) region.

low shares in the original pool with very few exceptions having high shares like Papua New Guinea (PNG) (almost 1.0) in PCRAFI or Jamaica (JAM) in CCRIF (0.94). After regional pooling, Papua New Guinea substantially lowers its risk share to 0.09, while Jamaica lowers it only to 0.60. Jamaica is also the country with the largest modeled losses within CCRIF. This large concentration of CCRIF's risk on a single country explains why the pool cannot exploit the full diversification potential within the region.

There are three possible Pareto optimal configurations of globally diversified PCRAFI and CCRIF (Fig. 3a). All these configurations have a higher diversification than the original pools and the regionally diversified original pools. This confirms that global pooling leads to a Pareto improvement of regionally diversified pools. The highest possible diversification is higher in PCRAFI (0.81, a 65% increase from its initial value) than in CCRIF (0.77, a 60% increase from its initial value). Although a trade-off exists in increasing risk diversification for the two pools, this does not seem to be relevant since the difference in risk diversification for the three possible globally diversified CCRIF pools ranges within 2 percentage points (from 0.75 to 0.77). Thus, only one configuration is selected for further exploration, namely the one leading to the highest diversification for PCRAFI.

For the selected configuration, the globally diversified PCRAFI pools a larger set of additional countries than the globally diversified CCRIF. Both PCRAFI and CCRIF pool many countries from their own region but PCRAFI, in addition, also pools many countries from LAC. Fewer countries are pooled from SSA and SA. Papua New Guinea (PNG) and Jamaica (JAM), namely the countries with the highest risk share in the original PCRAFI and CCRIF respectively, substantially decreased their risk share after global pooling. The same happens in the case of regional pooling. Unlike regional pooling, however, global pooling does not increase risk shares in any other country in the region. This happens because, in the globally diversified pools, the countries with the highest risk shares belong to another region and are poorly correlated with the countries in the original pool. In the globally diversified PCRAFI (EAP region), the countries with the highest share are Colombia (COL) (0.59) and Costa Rica (CRI) (0.33) in the LAC region, and Mauritius (MUS) (0.31) in the SSA region. In the globally diversified CCRIF (LAC region), the countries with the highest risk share are Malaysia (MYS) (0.42) and Viet Nam (VNM) (0.5) in the EAP region, and Bangladesh (BGD) (0.43) in the SA region.

## Discussion

Several international high-level policy agendas like the Sendai Framework[13] and the Paris Agreement[14] advocate for strengthening countries' financial resilience toward the impact of extreme natural hazards via *ex-ante* financial instruments. These instruments increase financial resilience because they guarantee a predictable flow of funding in the aftermath of disasters and thus allow governments to spread costs over time at a predictable rate.

The *InsuResilience* Global Partnership[15] identified sovereign catastrophe risk pools as a promising *ex-ante* disaster risk financing tool for low- and middle-income countries. Sovereign catastrophe risk pools represent a mechanism through which different countries pool their individual risk into a single diversified portfolio. Via risk diversification, risk pooling increases countries' financial resilience by either lowering countries' premiums to afford a given coverage or by increasing coverage for a given premium.

Risk diversification of currently existing pools, and therefore their members' financial resilience, may be limited because these pools were not designed with the primary goal of maximizing risk diversification and they pool risk only within regional borders. The present study addresses these two issues by introducing a method to find optimal risk pools, i.e., those with the highest risk diversification achieved with the least number of countries, and by applying it to assess the diversification potential of optimal global pooling.

The optimal pooling method is found to reasonably group countries by selecting those with low bilateral correlations or low risk contributions to the overall pool's risk. Optimal global pooling is found to increase risk diversification of all regional pools, to lower countries' shares in the pool's risk and to increase the number of countries that can profitably join the pool. Optimal global pooling, however, comes with trade-offs, as two or more pools need to pool the same set of countries to reach their highest possible diversification. This implies that multiple global groupings of countries are possible, and that no single grouping maximizes the diversification of all pools. In practice, this requires choosing the most desirable grouping among the many possible ones. Since risk pools require coordination, dialogue, and information sharing between participating countries, such a choice is not trivial and should rely on political considerations regarding which countries are more likely to cooperate successfully.

The method is also applied to explore whether risk diversification of two existing pools covering tropical cyclone risk, namely PCRAFI and CCRIF, would increase under optimal regional and global pooling. Overall, both optimal regional and global pooling increase risk diversification of the existing pools, implying that less capital would be required to insure these pools. This translates, in principle, into greater financial resilience. However, there are significant differences between results from regional and global pooling.

Optimal regional pooling allows PCRAFI to exploit the full diversification potential of its own region. The same is not true for CCRIF as its diversification is 11% lower than the maximum possible regional diversification. This implies a poor initial design of CCRIF in terms of only risk diversification criteria, likely due to CCRIF's overall loss profile being very concentrated on one single country's loss profile. Additional regional pooling cannot sufficiently reduce this initial high concentration on one single country.

Global optimal pooling offers greater potential for risk diversification than regional pooling as it provides a diversification of 65% to PCRAFI and 60% to CCRIF, both higher than the highest achievable regional diversifications. The trade-off relative to global pooling introduced above seems to be easily resolvable in this case since all global expansions of CCRIF provide very similar risk diversifications (within 2% points), which makes the selection of one single grouping less problematic.

These findings suggest that changes in the composition of CCRIF and PCRAFI via both optimal regional and global pooling can increase the pools' risk diversification. Although this could provide a higher coverage to member countries, and hence increase their financial resilience, it would not be sufficient on its own. The two pools are designed to merely provide sufficient coverage for a first response and countries often still rely on international aid to achieve a full recovery. Addressing this aspect would require a much more fundamental change in the pools' design than their composition.

The analysis in the present paper focused on tropical cyclone risk and therefore results cannot be generalized to other hazards. The method introduced is, however, general and can be applied to study optimal pools' compositions focusing on other hazards as well as multi-hazards. To expand the present work in the spirit of strengthening societal resilience against natural hazards, future research shall focus on assessing the potential effect of increasing risk diversification in the multi-hazard case, on the design of (re-)insurance policies, and on the composition of possible future optimal pools in light of socio-economic and climatic changes.

## Methods

The main benefit of risk pooling consists in lowering the capital requirements for risk coverage compared to when risks of the pool's members are covered independently. The more diversified the pool is, the higher the reduction in required capital. We first introduce a metric to quantify risk diversification, thus the extent of capital reduction,

and then describe the optimization problem to find optimal pools, namely the pools with the highest possible risk diversification achieved with the least number of countries.

## Risk diversification metric

Given a distribution of losses $L$ and a low enough threshold probability $\alpha$, one can define the *Value-at-Risk* at $\alpha$ (*VaR$_\alpha$*) for $L$ as the $\alpha$-*quantile of L*. *VaR* is widely used in the financial sector to determine the minimum capital requirements needed to compensate extreme losses from a portfolio, but it is has known limitations[20]. *VaR* tells nothing about the tail of the distribution, e.g., the magnitude of losses greater than *VaR$_\alpha$*, and it is not a coherent measure since it violates the sub-additivity property, implying that the portfolio's *VaR* may be higher than the sum of the portfolio's members' *VaRs*. An alternative metric is the *Conditional Value at Risk* (*CVaR*), also known as *Expected Shortfall* (*ES*). *ES* is a tail expectation measure, as it measures expected losses conditional on a loss higher than *VaR*, i.e., $ES_\alpha = E[L \mid L \geq VaR_\alpha]$. In addition, *ES* is a coherent measure since the *ES* of a portfolio is always equal to or greater than the sum of the portfolio's members' *ES*[21]. When dealing with portfolios, one can also define the *Marginal Expected Shortfall* (*MES*) of the $i^{th}$ portfolio's member as[22]:

$$MES_{\alpha_i} = E\left[L_i \mid L \geq VaR_\alpha\right] \tag{1}$$

where $L$ are the overall portfolio's losses, and $L_i$ are the portfolio's members' losses. *MES* indicates the countries' losses in the tail of the portfolio's loss distribution. Acharya et al.[22] show that the portfolio's *ES* can be defined as the sum of all *MES*:

$$ES_\alpha = E[L \mid L \geq VaR_\alpha] = \sum_i MES_{\alpha_i} = \sum_i E[L_i \mid L \geq VaR_\alpha] \tag{2}$$

Thus, the ratio between the portfolio's *ES* and the sum of the individual countries' *ES* indicates the degree of *Risk Concentration* (*RC*) of the pool:

$$RC = \frac{\sum_i E[L_i \mid L \geq VaR_\alpha]}{\sum_i E[L_i \mid L_i \geq VaR_{\alpha_i}]} \tag{3}$$

It follows from the additivity property of *ES* that *RC* is bounded between zero and one. An *RC* equal to one implies that all countries' tail losses contribute to the portfolio's tail losses, which makes risk pooling useless. This happens when all countries in the pool are perfectly correlated. *RC* goes to zero when only a small share of the countries' tail losses contributes to the portfolio's tail losses. Given *RC*, *Risk Diversification* (*RD*) can be defined as:

$$RD = 1 - RC = 1 - \frac{\sum_i E[L_i \mid L \geq VaR_\alpha]}{\sum_i E[L_i \mid L_i \geq VaR_{\alpha_i}]} \tag{4}$$

Finally, one can define the share, $s$, of an individual country's risk in the overall portfolio's risk as:

$$s_i = \frac{MES_i}{ES_i} = \frac{E[L_i \mid L \geq VaR_\alpha]}{E[L_i \mid L_i \geq VaR_{\alpha_i}]} \tag{5}$$

which could be used to derive fair premiums for countries in the pool.

## Optimal pools

As mentioned above, optimal pools are here defined as the pools with the highest possible diversifications achieved with the least number of countries. We find optimal pools via a two-step optimization. The first step aims at finding, given a set of countries, what subset allows achieving the maximum possible *RD, maxRD*. This subset, however, may be unnecessarily large since there are decreasing marginal

diversification benefits of adding new countries to a pool before a critical mass is reached[11]. Hence, some countries may have unnecessarily been added to the pool after the first optimization step. The second optimization step finds the smallest subset of countries within the previously found subset that still allows reaching *maxRD*.

We slightly modify the definition of *RD* provided above to account for the fact that countries may join different pools or not join a pool at all. Assuming a set of $n$ countries and $m$ possible pools a country may be part of, we define a vector $\boldsymbol{x}$ of length $n$ with integers from $0$ to $m$ that either allocates countries to one of the $m$ pools (values from $1$ to $m$) or indicates that no pool is joined (when equal to $0$). Then, we write the *RD* of the $j^{th}$ pool as:

$$RD_j(\boldsymbol{x}, j) = 1 - RC_j(\boldsymbol{x}, j) = 1 - \frac{\sum_i^n \mathbf{1}_j(x_i)E[L_i \mid L \geq VaR_\alpha]}{\sum_i^n \mathbf{1}_j(x_i)E[L_i \mid L_i \geq VaR_{\alpha,j}]} \tag{6}$$

Where $\mathbf{1}_j$ is the indicator function such that:

$$\mathbf{1}_j(x_i) = \begin{cases} 1 & x = j \\ 0 & x \neq j \end{cases} \tag{7}$$

In the first optimization step, for convenience and practical reasons, instead of maximizing *Risk Diversification* (*RD*) we minimize *Risk Concentration* (*RC*). The optimal allocation of countries, $\boldsymbol{x}^*$, which provides the minimum risk concentrations to the $m$ pools, $RC_1^*,...,RC_m^*$, can be found by solving the following $m$-objectives optimization problem:

$$\begin{aligned} \text{minimize} \quad & RC_1(\boldsymbol{x}, 1) \\ & \cdots \\ & RC_j(\boldsymbol{x}, j) \\ & \cdots \\ & RC_m(\boldsymbol{x}, m) \end{aligned} \tag{8}$$

The vector $\boldsymbol{x}^*$ indicates the set of the $n_1, ..., n_m$, countries that provide optimal diversifications in each of the $m$ pools.

The second optimization step requires solving a single-objective optimization for each of the $m$ pools. To do so, we define, for a given pool $j$, a binary vector $\boldsymbol{z}_j$ of length $n_j$ indicating which of the $n_j$ countries are still part of $j$ (when 1) or not (when 0). The smallest subset of countries within the set of $n_j$ countries which allows reaching the least concentration, $RC_j^*$, can then be found by solving:

$$\begin{aligned} \text{minimize} \quad & \sum_i^{n_j} z_{j,i} \\ \text{subject to} \quad & RC(z_j, 1) = RC_j^* \end{aligned} \tag{9}$$

The vector $z_j^*$ indicates the optimal set of countries for the pool $j$, namely the smallest set of countries that provide the highest maximum risk diversification.

Optimization is carried out via the python Pymoo package[23]. Pymoo provides a framework for solving single- and multi-objective optimization problems via state-of-art algorithms. We employ a basic genetic algorithm (GA) to solve the single objective optimizations and a unified non-dominated sorting genetic algorithm (U-NSGA-III) to solve the many-objective optimization problems. For these, we carried out a seed analysis and solved the optimization problem fifteen times. The final set of dominant solutions is then the dominant set across the fifteen sets of solutions so derived. Convergence plots of the two-step optimization are reported in Figs. S2–S7.

## Generation of tropical cyclone events

The historical record of hurricanes is too short for calculating *ES* for the 200-year event. Thus, a global synthetic tropical cyclone

track set containing over 90,000 events was generated for the historical period (between 1979 and 2019) based on the European Centre for Medium-Range Weather Forecasting (ECMWF)'s fifth-generation climate reanalysis dataset[24] using the model introduced by Emanuel et al.[23,25] and Emanuel et al.[26]. This model is based on a statistical-dynamical downscaling method. In detail, it propagates key statistical properties extracted from global reanalyses or climate models to generate a global, time-evolving, large-scale atmosphere-ocean environment. First, tropical cyclones are initiated using a random seeding technique where only the warm-core seed vortices in favourable environments for tropical cyclone formation survive and strengthen into tropical cyclones. These are then propagated via synthetic local winds using a beta-and-advection model. Finally, the tropical cyclone intensity along each track is simulated by a dynamical intensity model (CHIPS, Coupled Hurricane Intensity Prediction System)[26]. Note that the synthetic tropical cyclone event set frequency must be calibrated to match the observed number of events in the historical period.

A 10000-y time series is created using the synthetic datasets. To do so, we first used data from NOAA to identify—within the 1979–2019 period—those years characterized by persistent (more than 5) warm or cold seasons and those which are not. Then, we derived the frequencies of these year types within the considered period and used a multinomial distribution to generate a sequence of 10000-year types. Based on this sequence, 10000 years are sampled within the period 1979–2019. Following Emanuel et al. (2021)[27,28], a storm count is generated for each year by sampling from a Poisson distribution with lambda equal to the annual mean frequency of the events. Finally, for each year, we randomly sample from the whole event set as many events as the drawn storm count.

### The CLIMADA impact model

Damages from tropical cyclones are estimated using the open-source and -access CLIMADA impact model. As most weather and climate risk assessment models, damages in CLIMADA are assessed as a function of hazard, e.g., a tropical cyclone's wind field, exposure, e.g., the people and goods subject to such a hazard, and vulnerability, e.g., the degree to which hazard can harm exposure. Here we describe the specific CLIMADA set-up relative to the present study and refer the reader to Aznar-Siguan & Bresch[26,28] and Bresch & Aznar-Siguan[29] for a more detailed description of CLIMADA.

Tropical cyclone hazard modeling in CLIMADA is based on a parametric wind model following Holland[30], which is run on each synthetic tropical cyclone track. The wind model computes the gridded 1-min sustained winds at 10 m above the ground as the sum of a circular wind field and the translational wind speed that arises from the tropical cyclone movement. For this study, we calculate wind fields at a resolution of 300 arc-sec (~10 km).

Exposure for all considered countries is modeled via the LitPop approach proposed by Eberenz et al.[31]. LitPop is a globally consistent methodology to disaggregate asset value data proportional to a combination of nightlight intensity and geographical population data. Vulnerability relates hazard intensity with the percentage of exposure damage. We use the vulnerability functions generated by Eberenz et al.[32] which were calibrated on tropical cyclone damages for various regions around the world.

### Data availability

The synthetic TC data are property of WindRiskTech L.L.C., which is a company that provides hurricane risk assessments to clients worldwide. Upon request, the company provides datasets free of charge to scientific researchers, subject to a non-redistribution agreement. The TC data are fed into CLIMADA to calculate TC impacts. The data so derived are available at https://doi.org/10.5281/zenodo.7371742[33].

### Code availability

The source code to reproduce all results in the present paper is available at https://doi.org/10.5281/zenodo.7371742[33].

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

## Acknowledgements

We acknowledge Kerry Emanuel for generating the tropical cyclone data and providing comments on an early version of the manuscript. A.C. was funded by the EU Horizon 2020 project Remote Climate Effects and their Impact on European sustainability, Policy and Trade (RECEIPT), Grant agreement no. 820712.

## Author contributions

A.C. and E.S. conceived and designed the research. A.C. carried out the research and wrote the manuscript. S.M. processed part of the data and wrote part of the method section. A.C., E.S., O.M., D.N.B. analysed the results. All authors (A.C, E.S., S.M., O.M., D.N.B.) reviewed and edited the manuscript.

## Competing interests

The authors declare no competing interests.
