## [Peer Review File · Nature Communications]

Increasing countries' financial resilience through global catastrophe risk poolingReviewers' Comments:

Reviewer #1:

Remarks to the Author:

My research background allows me to provide comments on some of the broader aspects regarding this submission - whereas I am not qualified to comment on the submission's methodological aspects (i.e. the insurance mathematical aspects).

* I believe that the submission would need to undergo a linguistic check. This first of all applies to the presentation of the methods where the text appear to be incomplete (see for example lines 462, 478, 489, 492, 498 and 502). However, there are also instances where there appear to be more 'traditional' linguistic problems (for example lines 30, 32, 83, 148).

* In lines 105-107 the authors observe that they apply the World Bank's official regional classification. However, they do not provide a reference to where this classification may be found. With regards to Africa, I am aware of the World Bank referring to Sub-Saharan Africa as a regional group (see: <https://datahelpdesk.worldbank.org/knowledgebase/articles/906519>). However, I am not aware of the World Bank's definition of the regional group "South-East Africa" encompassing Somalia and Ethiopia. It would be useful if the authors could provide a reference in this respect.

* My understanding of the authors' primary objective is that they want to clarify whether increased geographic distribution of the members of catastrophe risk pooling schemes (with particular regard to tropical cyclones) will lead to enhanced risk diversification (vis-à-vis regional risk pools). Since a tropical cyclone will have a geographically limited impact, there would seem to be a strong presumption that the examination will show there to be enhanced risk diversification (as indeed the authors' examination does). It therefore seems to me that the principal merit of the examination is to confirm that the immediate presumption is correct.

* The authors appear to me to focus exclusively upon the 'insurance mathematical' aspects of the risk pools. This may be well-founded. However, I personally would have liked to see that the authors are aware that in practice the existing risk pools have turned out to be suffering from a number of weaknesses - and that these weaknesses will not be rectified simply by giving these pools a broader geographic coverage.

* The authors leave an impression that the members of the different risk pools can shoulder the costs of the pools themselves - and thereby these members will not be dependent upon assistance from donors when a cyclone hits. I believe that it may be useful to note that at least two of the three risk pool schemes are not fully paid by the member countries (i.e. these schemes are also based on donor contributions), and that in any case all the schemes merely provide an important 'first response' (which must be followed by additional assistance - where donors will play an important part).

* Finally, the authors (correctly) point out that the main objective behind the existing pools has been to serve the interests of the individual members (i.e. the member countries) - and that risk diversification was merely seen as an instrument towards this objective (lines 86-87). In light of this and in light of the fact that the authors' finding (that enhanced geographical spread will lead to better risk diversification) is so obvious that, presumably, the member countries are already aware thereof, it would be useful if the authors would consider whether their finding is likely to prompt these members to change the existing member composition of the pools.

Morten Broberg

Reviewer #2:

Remarks to the Author:

I appreciated the opportunity to review this very compelling paper comparing optimal forms of pooling risk between regional and global collections of climate change vulnerable country sets. I have often wondered how well the current regional sovereign risk facilities (CCRIF, ARC, PCRAFI) are at diversifying risk, and this paper provides convincing evidence that the answer is, far from as good as they could be!

I do not see the need for any major changes in terms of approach and argumentation, and I think this paper very much deserves to be published in Nature Communications. I did have more minor quibbles with how the authors shuttle back and forth between text and graphs. There is so much going on in each collection of graphs, I do not think the authors do the readers and favors by pointing them to parts of the collection using phrases such as (lines 129): "(first column from the second to last row in Figure 1)." Perhaps this can be cleared up by simply labeling each panel graph in the figures with a lower case letter and using the letters to direct the readers within the figure to the panels, or subparts of the panels, they want the reader to focus on. Might provide for a smoother experience.

Line 110: "pools for short in the aftermath" -- maybe the authors mean "pools for short hereinafter"? i.e. following this parenthetical comment?

Also, there is a hint at the end of the paper (lines 309-3011), about institutional features that enable effective risk sharing. This deserves a whole other papers, but I appreciated the nod to the social components of how risk operates, which are perhaps less subject to optimization. There is also an imaginative component to how risk might be pooled, as well, which is that those at risk must see themselves as belonging to what Ulrich Beck called a shared "community of fate." These are factors that are likely upstream of political considerations, but no less important in terms of enabling cooperation and legitimacy to these ex ante risk transfer efforts between the vulnerable countries.

Overall, bravo on an important piece of research that I hope will help shape the evolution of these instruments and improve their penetration.

We thank the reviewers for their valuable feedback, which we believe helped us in improving our manuscript. Below we provide a point-by-point reply to each comment.

Reviewer #1:

My research background allows me to provide comments on some of the broader aspects regarding this submission - whereas I am not qualified to comment on the submission's methodological aspects (i.e. the insurance mathematical aspects).

* I believe that the submission would need to undergo a linguistic check. This first of all applies to the presentation of the methods where the text appear to be incomplete (see for example lines 462, 478, 489, 492, 498 and 502).

The reviewer is right that some text appeared to be incomplete. This was due to a wrong citation style. All erroneous citations have been fixed and the text no longer looks incomplete.

However, there are also instances where there appear to be more 'traditional' linguistic problems (for example lines 30, 32, 83, 148).

The overall manuscript underwent thorough language checks.

* In lines 105-107 the authors observe that they apply the World Bank's official regional classification. However, they do not provide a reference to where this classification may be found. With regards to Africa, I am aware of the World Bank referring to Sub-Saharan Africa as a regional group (see: <https://datahelpdesk.worldbank.org/knowledgebase/articles/906519>). However, I am not aware of the World Bank's definition of the regional group "South-East Africa" encompassing Somalia and Ethiopia. It would be useful if the authors could provide a reference in this respect.

The reviewer indeed points to the same regional classification we use in the manuscript (the link is now cited at line 125 of the ms with tracked changes). Originally, we named some regions slightly differently because we wanted the geographical names to better relate to areas of tropical cyclones activity. We agree with the reviewer that consistency with the original naming is important to avoid confusion, and we therefore renamed the "South-East Africa" region into "Sub-Saharan Africa" and the "Central America & Caribbean" region into

“Latin America & Caribbean” throughout the manuscript to guarantee consistency with the World Bank’s regional classification.

* My understanding of the authors' primary objective is that they want to clarify whether increased geographic distribution of the members of catastrophe risk pooling schemes (with particular regard to tropical cyclones) will lead to enhanced risk diversification (vis-à-vis regional risk pools). Since a tropical cyclone will have a geographically limited impact, there would seem to be a strong presumption that the examination will show there to be enhanced risk diversification (as indeed the authors' examination does). It therefore seems to me that the principal merit of the examination is to confirm that the immediate presumption is correct.

The reviewer is right that we hypothesized an increase in risk diversification with global pooling. This hypothesis, however, had still to be tested and – most importantly – the extent of such increase had to be quantified. Furthermore, our main goal is methodological, and the method proposed was applied to investigate not only global but also regional expansions of the pools. We could show that the existing pools are suboptimal, and that they might reach a higher diversification even by pooling within their geographical boundaries.

* The authors appear to me to focus exclusively upon the 'insurance mathematical' aspects of the risk pools. This may be well-founded. However, I personally would have liked to see that the authors are aware that in practice the existing risk pools have turned out to be suffering from a number of weaknesses - and that these weaknesses will not be rectified simply by giving these pools a broader geographic coverage.

We agree that we are tackling only one of the many limitations of existing sovereign catastrophe pools. We now make it clearer at lines 93-100 of the ms with tracked changes.

* The authors leave an impression that the members of the different risk pools can shoulder the costs of the pools themselves - and thereby these members will not be dependent upon assistance from donors when a cyclone hits. I believe that it may be useful to note that at least two of the three risk pool schemes are not fully paid by the member countries (i.e. these schemes are also based on donor contributions), and that in any case all the schemes merely provide an important 'first response' (which must be followed by additional assistance - where donors will play an important part).

We agree and we extended the text in the introduction (lines 93-100) and discussion (lines 376-382) sections to account for these aspects.

* Finally, the authors (correctly) point out that the main objective behind the existing pools has been to serve the interests of the individual members (i.e. the member countries) - and that risk diversification was merely seen as an instrument towards this objective (lines 86-87). In light of this and in light of the fact that the authors' finding (that enhanced geographical spread will lead to better risk diversification) is so obvious that, presumably, the member countries are already aware thereof, it would be useful if the authors would consider whether their finding is likely to prompt these members to change the existing member composition of the pools.

Our results indeed suggest that a change in the compositions of the pools could lead to a better diversification (see e.g., lines 361-374 of the manuscript with tracked changes). Establishing whether this is practically feasible, however, goes beyond science and the scope of this paper.

Reviewer #2

I appreciated the opportunity to review this very compelling paper comparing optimal forms of pooling risk between regional and global collections of climate change vulnerable country sets. I have often wondered how well the current regional sovereign risk facilities (CCRIF, ARC, PCRAFI) are at diversifying risk, and this paper provides convincing evidence that the answer is, far from as good as they could be!

I do not see the need for any major changes in terms of approach and argumentation, and I think this paper very much deserves to be published in Nature Communications. I did have more minor quibbles with how the authors shuttle back and forth between text and graphs.

We thank the reviewer for appreciating our work. We addressed both the textual and graphical comments.

There is so much going on in each collection of graphs, I do not think the authors do the readers and favors by pointing them to parts of the collection using phrases such as (lines 129): "(first column from the second to last row in

Figure 1)." Perhaps this can be cleared up by simply labeling each panel graph in the figures with a lower case letter and using the letters to direct the readers within the figure to the panels, or subparts of the panels, they want the reader to focus on. Might provide for a smoother experience.

The suggestion has been implemented. A letter was added to each panel and all descriptions were changed accordingly.

Line 110: "pools for short in the aftermath" -- maybe the authors mean "pools for short hereinafter"? i.e. following this parenthetical comment?

The text was changed at the suggested line and elsewhere where appropriate.

Also, there is a hint at the end of the paper (lines 309-3011), about institutional features that enable effective risk sharing. This deserves a whole other papers, but I appreciated the nod to the social components of how risk operates, which are perhaps less subject to optimization. There is also an imaginative component to how risk might be pooled, as well, which is that those at risk must see themselves as belonging to what Ulrich Beck called a shared "community of fate." These are factors that are likely upstream of political considerations, but no less important in terms of enabling cooperation and legitimacy to these ex ante risk transfer efforts between the vulnerable countries.

We agree with the reviewer, this are very relevant aspects. Also, as they rightly point out, these would deserve a paper on their own and we decided not to elaborate about this aspect further in our manuscript.

Overall, bravo on an important piece of research that I hope will help shape the evolution of these instruments and improve their penetration.